# Cyberbullying and Obesity in Adolescents: Prevalence and Associations in Seven European Countries of the EU NET ADB Survey

**DOI:** 10.3390/children8030235

**Published:** 2021-03-18

**Authors:** Theodoros N. Sergentanis, Sofia D. Bampalitsa, Paraskevi Theofilou, Eleni Panagouli, Elpis Vlachopapadopoulou, Stefanos Michalacos, Alexandros Gryparis, Loretta Thomaidis, Theodora Psaltopoulou, Maria Tsolia, Flora Bacopoulou, Artemis Tsitsika

**Affiliations:** 1MSc Program “Strategies of Developmental and Adolescent Health”, 2nd Department of Pediatrics, “P. & A. Kyriakou” Children’s Hospital, School of Medicine, National and Kapodistrian University of Athens, 115 27 Athens, Greece; tsergentanis@yahoo.gr (T.N.S.); bampalitsa.sofia@gmail.com (S.D.B.); elenpana@med.uoa.gr (E.P.); al.grip@gmail.com (A.G.); dr_thomaidis@yahoo.gr (L.T.); tpsaltop@med.uoa.gr (T.P.); mtsolia@med.uoa.gr (M.T.); 2Department of Clinical Therapeutics, “Alexandra” Hospital, School of Medicine, National and Kapodistrian University of Athens, 115 28 Athens, Greece; 3General Management of Health Services, Ministry of Health, 101 87 Athens, Greece; pardrothe@gmail.com; 4Department of Endocrinology-Growth and Development, “P. & A. Kyriakou” Children’s Hospital, 115 27 Athens, Greece; elpis.vl@gmail.com (E.V.); stmichalakos@gmail.com (S.M.); 5Center for Adolescent Medicine and UNESCO Chair Adolescent Health Care, First Department of Pediatrics, “Agia Sophia” Children’s Hospital, School of Medicine, National and Kapodistrian University of Athens, 115 27 Athens, Greece; bacopouf@hotmail.com

**Keywords:** cyberbullying, overweight, obesity, adolescents, mental health

## Abstract

Background: overweight and obese individuals may often face aggressive messages or comments on the internet. This study attempts to evaluate the association between cyberbullying victimization and overweight/obesity in adolescents participating in the European Network for Addictive Behavior (EU NET ADB) survey. Methods: a school-based cross-sectional study of adolescents aged 14–17.9 years was conducted (*n* = 8785) within the EU NET ADB survey, including data from seven European countries (Germany, Greece, Iceland, the Netherlands, Romania, Poland, Spain). Complex samples and univariate and multivariate logistic regression analyses were performed. Results: overall, overweight adolescents were more likely to have been cyberbullied compared to their normal weight peers (adjusted OR (Odds ratio) = 1.20, CI (confidence intervals): 1.01–1.42); this association was pronounced in Germany (adjusted OR = 1.58, CI: 1.11–2.25). In Iceland, obese adolescents reported cyberbullying victimization more frequently compared to their normal weight peers (adjusted OR = 2.87, 95% CI: 1.00–8.19). No significant associations with cyberbullying victimization were identified either for obese or overweight adolescents in Greece, Spain, Romania, Poland, and the Netherlands. Conclusions: this study reveals an overall association between cyberbullying victimization and overweight on the basis of a sizable, representative sample of adolescent population from seven European countries. Country-specific differences might reflect differential behavioral perceptions, but also normalization aspects.

## 1. Introduction

Obesity is a major public health issue worldwide [1]. The prevalence of childhood and adolescent obesity has tripled between 1980 and 2000 in the United States [2]; accordingly, various European countries have faced an increase in childhood and adolescence obesity rates [3]. Childhood and adolescent obesity has been associated with various health conditions, including cardiovascular, endocrine, musculoskeletal, and gastrointestinal disease [4], but also with psychosocial conditions, including internalizing and externalizing problems [5] and problematic internet use [6]. 

Prevalence rates of cyberbullying vary considerably across studies, reflecting heterogeneity in definitions of cyberbullying, occurrence period used, frequency of cyberbullying events, and demographic features of the study sample [7,8,9,10,11]. In 2017, a scoping review [7] of 159 prevalence studies, determined that online victimization rates within the last year varied widely between 1.0% and 61.1%, whereas lifetime cyberbullying victimization rates ranged between 4.9% and 65.0%. A European cross-national study within the DAPHNE program in 2008–2009 estimated the prevalence of cyberbullying to 6.2% in Spain, 4% in the United Kingdom, and 5.4% in Italy [8]. In the "EU Kids Online" study [9] in 2010, the estimated prevalence rate was between 2 and 14%, whereas the “Net Children Go Mobile” program reported an increase from 7% to 12% from 2010 to 2014 [10]. In the European Network for Addictive Behavior (EU NET ADB) cross-sectional school-based study, among seven European countries the highest rate of cybervictimization has been found in Romania (37.3%) and the lowest in Spain (13.3%) [11]. 

Overweight and obese youths are more likely to face bullying than their non-obese peers [12], according to a meta-analysis [13]. In the online world, overweight and obese individuals may often face aggressive messages or comments on the internet; a systematic assessment of YouTube comments has revealed that weight stigma can “go viral” on the internet, attacking overweight men and women [14]. In light of the above, this study attempts to evaluate the association between cyberbullying victimization and overweight/obesity in adolescents participating in the European Network for Addictive Behavior (EU NET ADB) survey. 

## 2. Materials and Methods

### 2.1. Study Design

This study is part of the larger research EU NET ADB survey, which was performed in adolescents from seven countries: Greece, Spain, Poland, Germany, the Netherlands, Romania, and Iceland. The respective ethical committees of participating countries approved the study protocol. A school-based random clustered probability sample was drawn in each country; school class (ninth and tenth grade) was set as the primary sampling unit. The sampling frames were the official national lists, stratified by region and population density; one hundred classes (2000 students) per country were randomly selected, as previously described in detail [11]. During the period from October 2011 to May 2012. [15]. The research included students from all the selected classes, which were present in class on the data collection day; signed parental consent was necessary for enrolment in the study. The students completed the anonymous questionnaires during the school hours. Participation rate was equal to 85% of students on the class registers, namely 13,708 adolescents. Absence of data on age and gender led to the exclusion of 424 participants. Thus 13,284 adolescents (female/male 7000/6284, age between 14 and 18 years old, mean age of first internet use 9.6 years) participated in this study. 

All of the children selected for the study attended middle school, specifically the 9th and 10th grades. All of the students from each selected class participated in the procedure, so that every student had an equal opportunity to participate in the study. The national student registry for each school performed the random selection. Each class contributed to the study with 20–25 students with the researchers recording information on the class size, parental consent, and the absent students on the day of the research. 

The study researchers provided schools with self-reporting questionnaire that consisted of 52 items including questions pertaining to personal characteristics, family status, and internet use, during the period between October 2011 and May 2012, on printed forms and was answered during the school hours. The researchers and teachers were present during the procedure according to the local school rules. Self-report questionnaire guarantee confidentiality and reduce the risk of biased reporting. Before the start of the procedure, researchers informed the adolescents of the nature of the study and its goals, and gave information on the questionnaires, with instructions on how to answer them. The adolescents were encouraged to ask questions. Researchers clarified that they would answer general questions to the whole classroom and afterwards answer personal questions of the students in private. Students could abandon the process at any time. The whole process lasted an hour and adolescents with special needs (i.e., learning difficulties) answered the questionnaire with the help of the teachers. The sampling and data collection procedures are described in detail in Tsitsika et al. [15].

In this study, body weight and height were self-reported by the participating adolescents. Body mass index (BMI, kg/m^2^) was used to determine childhood overweight and obesity. Overweight was defined as a BMI ≥ 85th percentile and <95th percentile; obesity was defined as a BMI ≥ 95th percentile for specific age and sex, according to Cole et al. [16,17]. 

The definition of cyberbullying victimization in the study was the following: “Sometimes children or teenagers can do hurtful or nasty things to someone and this can often be quite a few times on different days over a period of time, for example. This can include: teasing someone in a way this person does not like; spreading false/malicious rumors; sending someone mean or threatening messages; systematically excluding, ignoring, and isolating. When people are hurtful or nasty to someone in this way, it can happen on the internet (e-mail, instant messaging, social networking, chat rooms)” [11]. Participants were asked to report if they were victims of such a hurtful or nasty incident in the past 12 months on the internet and possible responses were “no”, “yes”, and “do not know/prefer not to say”. The question was adapted from the “EU Kids Online” survey [18], as described previously [11].

Since the current study focused on the association between obesity/overweight in adolescents and cyberbullying victimization, underweight adolescents were excluded. Thus, this statistical analysis included 8785 adolescents, aged from 14 to 17.9 years old. 

### 2.2. Statistical Analysis

The statistical analysis of the present work included both descriptive and inferential statistical processing. Descriptive statistics were presented for both the entire sample size and each country separately. Categorical variables are presented as absolute and relative (%) frequencies. Complex samples and univariate and multivariate logistic regression analyses were performed to evaluate the relationship between cyberbullying victimization (dependent variable), overweight and obesity (versus normal weight as reference categories); in the multivariate analysis, gender, parental marital status, and age in categories were entered as covariates. A joint analysis merging overweight/obesity (versus normal weight) was also performed. Complex samples modeling was implemented to appropriately estimate standard errors and confidence intervals, reflecting the multistage design of this survey; in the complex samples approach, countries were treated as strata and school classes as clusters. Odds ratios (OR) and 95% confidence intervals (Cl) were estimated. The level of statistical significance was set at 0.05. Statistical analysis was performed using the complex samples routine in SPSS Version 25.0 (IBM Corp. Armonk, NY, USA). 

## 3. Results

This study analyzed 8785 adolescents from the EU NET ADB survey, aged 14–17.9 years old. Regarding participants by country, 17.2% (*n* = 1516) of adolescents were from Greece, 14.7% (*n* = 1295) from Spain, 14.1% (*n* = 1240) from Romania, 13.5% (*n* = 1190) from Poland, 18.4% (*n* = 1621) from Germany, 8.3% (*n* = 727) from the Netherlands, and 13.6% (*n* = 1196) from Iceland. 

Greece had the highest prevalence of overweight/obesity combined (21.2%), followed by Germany (15.7%), Iceland (12.8%), Poland (12.2%), Spain (11.7%), Romania (10.6%), and the Netherlands (7.6%). Table 1 shows the adolescent BMI categories in each country.

Romania had the highest prevalence of cyberbullying victimization (38.1%), followed in descending order by Greece (27.1%), Germany (25.5%), Poland (22.4%), the Netherlands (14.5%), Iceland (14.2%), and Spain (13.0%)

In the overall results, analyzing all participating seven countries jointly, overweight adolescents were more likely to have been cyberbullied compared to their normal weight peers (adjusted OR = 1.20, 95% CI: 1.01–1.42); on the other hand, no association with obesity emerged. Table 2 presents the results of the complex samples logistic regression analysis.

In country-specific associations, different patterns arose. In Iceland, obese adolescents reported having been cyberbullied more frequently compared to their normal weight peers both in the univariate (crude OR = 2.73, 95% CI: 1.21–6.19) and the multivariate analysis (adjusted OR = 2.87, 95% CI: 1.00–8.19) whereas no association with overweight was noted. In Germany, the multivariate analysis revealed a significant association between cyberbullying victimization and overweight (adjusted OR = 1.58, 95% CI: 1.11–2.25), whereas no association with obesity emerged; the association with overweight/obesity pooled together, reached significance (adjusted OR = 1.41, 95% CI: 1.00–1.98), replicating the finding in overweight subjects. No significant associations were identified either for obese or overweight adolescents in Greece, Spain, Romania, Poland, and the Netherlands.

Males reported less frequently to have been cyberbullied compared to females (adjusted OR = 0.75, 95% CI: 0.67–0.84); the association was replicated in Iceland, the Netherlands and Spain. On the other hand, older adolescents (16–17.9 years old) were more often cyberbullied compared to younger adolescents (14–15.9 years old; adjusted OR = 1.25, 95% CI: 1.11–1.39). Adolescents with divorced/separated parents were more often victims of cyberbullying compared to their peers with married/living together parents (adjusted OR = 1.19, 95% CI: 1.06–1.35, Table 3); the association was replicated in Greece and Iceland.

## 4. Discussion

This study reveals original associations between cyberbullying victimization and overweight/obesity on the basis of a sizable, representative sample of adolescent population from seven European countries. The association between overweight and cyberbullying victimization was prominent in Germany, whereas an association with obesity emerged in Iceland.

Comparing the present results with the previously published literature, it should be acknowledged that very few studies have examined the associations between cyberbullying and obesity. One study was the Health Behavior in School-aged Children (HBSC) 2005/2006 U.S. study [19], in which 7508 adolescents participated and no association was found between cybervictimization and weight group. Another pioneering study was the epidemiological study conducted in the U.S. youth weight loss camps, which showed that more than half of the adolescent participants were cyberbullied. Almost 61% reported experiencing mean or embarrassing posts online; cyberbullying had occurred in similar percentages among previously and currently obese adolescents [20]. A cross-sectional study conducted in Belgium [21] included 102 obese adolescents in a treatment facility for severely obese and compared them with an equal number of normal-weight youngsters from the HBSC study of the Flemish adolescent population; the study showed that severely obese adolescents were cyberbullied at least once during the last six months in greater percentage than their normal weight peers (17.2% and 7.8%, respectively). Interestingly, the results of this research are different from the HBSC U.S. study [19] that found no association between cyberbullying and obesity.

In our study, country specific analyses showed an association between cyberbullying victimization and overweight/obese adolescents in some countries, whereas in other countries no associations were documented.

In Iceland, cyberbullying victimization was significantly associated with obesity in the current study. Although there are no previous peer-reviewed published data from Iceland about this association, there is research on traditional bullying. A cross-sectional survey of adults, some of them being residents of Iceland, showed that weight status was the first cause of bullying in the same regions [22].

Similarly, in Germany, there was a positive association between cyberbullying victimization and overweight. Although it would be reasonable to presume that cyberbullying victimization might be more severe in the obesity group than in the overweight group, this was not documented in the aforementioned country; nevertheless, the small number of German adolescents with obesity (*n* = 46) may account for the lack of anticipated association. The analysis, pooling together overweight and obesity, however, replicated the association between excess body weight and being a victim of cyberbullying in Germany. 

As far as Greece and Spain are concerned, there was no association between cyberbullying victimization and overweight/obesity. It would be tempting to hypothesize a “normalization” of overweight/obesity among adolescents in these Southern European countries, as they rank in the top ranks of obesity prevalence in various surveys, including the most recent report of the World Health Organization (WHO) European Childhood Obesity Surveillance Initiative (COSI) [23]. Indeed, such a pattern has been also reported in the United Kingdom, where the upward trend in underassessment of overweight and obesity status (37% to 40% in men; 17% to 19% in women between 1997 and 2015) is possibly a result of the normalization of overweight and obesity [24]. Regarding the studied associations, there is scarcity of data in Greece; however, in Spain, a study including 676 adolescents among several cities, associated cyberbullying with eating disorders and overweight preoccupation [25]. In addition, a recent study in 3145 adolescents in Asturias (Spain) showed that adolescents perceiving themselves as overweight were more often victims of bullying offline or online than those not perceiving themselves as overweight [26]. 

In our study, there were no associations between cyberbullying and overweight/obesity in Romania, Poland and the Netherlands. In the Netherlands, however, a previous study, measuring BMI and teacher-reported bullying behavior among 4364 children highlighted that at school, a high BMI correlated with victimization, with obese children being more likely victims but also aggressors. [27].

A connection between cyberbullying and traditional bulling has been reported, after more than a decade of research. In an anonymous web-based survey of 1454 adolescents in the U.S., 72% reported at least one instance of cyberbullying; 85% of the latter also experienced bullying in school [28]. In another study conducted in the U.S., results showed that students’ roles (victims or perpetrators) in traditional bullying predicted the same role in cyberbullying, [29]. In an early survey of two epidemiological studies, cyberbullying was less frequent than traditional bullying, but occurred at an appreciable extent [30]. Nevertheless, it is also clear that, although linked, cyberbullying and traditional bullying have their own distinct features, as a consequence of the means used [31]; importantly, the implications of cyberbullying in terms of affecting psychosocial health may be worse than traditional bullying [32].

Regarding the underlying reasons for the association between overweight/obesity and cyberbullying sociological theories attempt to explain discrimination related to specific human characteristics, such as obesity. The opinion that overweight/obesity is a personal responsibility, providing a negative cultural feedback, is perceived in anti- fat behaviors mostly in individualist cultures comparing to collectivist cultures [33]. Weight stigma has been associated with stereotypical beliefs linking adolescent obesity with laziness and self-indulgence [34]; importantly, children and adolescents seem considerably influenced by stereotypes based on physical cues [35]. Attributions of behavioral causes of obesity and beliefs that obesity occurs due to lack of willpower and personal responsibility have been associated with stronger prejudice against people with excess body weight, termed as “weight bias” [36]. Beliefs supporting that the cause of obesity is related to factors pertaining to personal control, such as overeating or a sedentary lifestyle, have been associated with weight bias; on the contrary, beliefs attributing obesity to factors that lie outside of personal control, such as genes and environmental parameters, attenuate prejudice [37,38]. Moreover, adolescent obesity, traditional bullying and cyberbullying share common correlates, such as low self-esteem, feelings of loneliness, and psychosocial difficulties, which could enhance a vicious circle of reciprocal interactions perpetuating the phenomenon [35]. 

As far as covariates associated with cyberbullying are concerned, there is variety of findings regarding the role of gender. Boys have been reported more likely to be categorized as perpetrators, victims or victim-perpetrators compared to girls [39,40,41]. whereas other studies have not found statistically significant differences for cyberbullying among boys and girls, either as victims or perpetrators [40]. Gender differences might also pertain to the means of cyberbullying. For example, females seem to be victimized through e-mails more often than males [40], but males are bullied through text messages more often than females [41]. Regarding age, studies of traditional bullying have shown that the prevalence of bullying peaks in adolescence, when youth tries to dominate within the social hierarchy [42,43]. The association between being a victim of cyberbullying and separated/divorced parental marital status is in line with a recent systematic review that highlighted that students from single-parent households, divorced/widowed parents were more likely to be cyberbullied [44]. This study is part of the larger research EU NET ADB survey; other correlates of cybervictimization in the EU NET ADB survey include time spent on social network sites and time spent online, internalizing and externalizing problems [11]. 

Despite its originality, this study bears certain limitations. First, adolescents not attending school or being absent on the day of data collection could not be included in this survey. Second, this was a cross-sectional study, and direction of associations could not be indicated. The survey was based on self-report data, hampered by recall bias; this limitation pertained not only to cyberbullying victimization but also weight and height. Moreover, the study was conducted in seven European countries; the results cannot be used to generalize the outcome for Europe as a whole, or to other parts of the world. In addition, the subgroup of overweight adolescents was much larger than that of obese adolescents, a fact that may have limited the precision of effect estimates in the latter subgroup. Furthermore, data about personality traits and details about interpersonal relationships, that could represent meaningful confounders, were not available to be adjusted for in the multivariate models. Potential alternative explanations underlying the observed associations between overweight/obesity and cyberbullying victimization cannot be precluded, since genetic characteristics, dietary intakes and motivation for physical activity were not examined in this survey.

As a whole, this study focused on the online environment, examining cyberbullying victimization and not the associations with traditional bullying. Moreover, this study was conducted in 2011–2012. More recent studies should evaluate whether the associations persist in the 2020s; however, a major strength of this study was the large sample of adolescents from randomly selected schools, across seven participating European countries, for which previous research at a national representative sample had not been published. 

## 5. Conclusions

Our findings indicated that cyberbullying is associated with overweight and obesity among adolescents in some countries. There is a need for further investigation using longitudinal designs. The present findings are also interesting from a preventive and therapeutic perspective; schools can apply interventions and take measures in order to prevent cyberbullying towards overweight and obese adolescents, such as application of social emotional learning (SEL)-focused programs [45,46].

## Figures and Tables

**Table 1 children-08-00235-t001:** Description of the study sample (*n* = 8785). Frequency of normal weight, overweight and obese adolescents in the seven countries participating in the European Network for Addictive Behavior (EU NET ADB) survey.

		Frequency	Percentage
**Iceland**	Normal	1043	87.2
	Overweight	127	10.6
	Obese	26	2.2
**Netherlands**	Normal	672	92.4
	Overweight	46	6.3
	Obese	9	1.2
**Germany**	Normal	1366	84.3
	Overweight	209	12.9
	Obese	46	2.8
**Poland**	Normal	1045	87.8
	Overweight	122	10.3
	Obese	23	1.9
**Romania**	Normal	1108	89.4
	Overweight	126	10.2
	Obese	6	0.5
**Spain**	Normal	1143	88.3
	Overweight	137	10.6
	Obese	15	1.2
**Greece**	Normal	1194	78.8
	Overweight	273	18.0
	Obese	49	3.2

**Table 2 children-08-00235-t002:** Results of complex samples and univariate and multivariate logistic regression analyses for the association between cyberbullying victimization and body mass index (BMI) status (obese, overweight, overweight/obese vs. normal weight), overall and by country.

	Categories	Univariate OR (95% CI)	Multivariate OR (95% CI) *
Overall results	Overweight vs. normal	1.11 (0.94–1.30)	1.20 (1.01–1.42) §
	Obese vs. normal	0.99 (0.67–1.48)	1.00 (0.64–1.41) §
	Overweight/obese vs. normal	1.10 (0.94–1.28)	1.17 (0.99–1.38)
Results by country			
**Greece**	Overweight vs. normal	0.92 (0.67–1.27)	0.94 (0.67–1.33)
	Obese vs. normal	1.09 (0.55–2.17)	0.98 (0.45–2.15)
	Overweight/obese vs. normal	0.95 (0.70–1.30)	0.95 (0.69–1.33)
**Spain**			
	Overweight vs. normal	1.03 (0.60–1.77)	1.11 (0.63–1.97)
	Obese vs. normal	1.05 (0.23–4.77)	1.31 (0.27–6.22)
	Overweight/obese vs. normal	1.04 (0.6–1.70)	1.14 (0.68–1.90)
**Romania**			
	Overweight vs. normal	1.16 (0.82–1.66)	1.07 (0.75–1.54)
	Obese vs. normal	0.32 (0.03–2.75)	0.28 (0.03–2.48)
	Overweight/obese vs. normal	1.11 (0.78–1.58)	1.03 (0.72–1.47)
**Poland**			
	Overweight vs. normal	0.74 (0.47–1.18)	0.89 (0.54–1.47)
	Obese vs. normal	0.54 (0.18–1.59)	0.39 (0.09–1.61)
	Overweight/obese vs. normal	0.79 (0.49–1.27)	0.81 (0.51–1.30)
**Germany**			
	Overweight vs. normal	1.20 (0.90–1.60)	1.58 (1.11–2.25)
	Obese vs. normal	0.99 (0.53–1.86)	0.76 (0.31–1.84)
	Overweight/obese vs. normal	1.25 (0.91–1.73)	1.41 (1.00–1.98)
**Netherlands**			
	Overweight vs. normal	1.16 (0.54–2.48)	1.51 (0.66–3.44)
	Obese vs. normal	0.78 (0.08–6.90)	1.22 (0.13–11.41)
	Overweight/obese vs. normal	1.22 (0.57–2.62)	1.47 (0.66–3.30)
**Iceland**			
	Overweight vs. normal	0.83 (0.51–1.37)	0.99 (0.56–1.74)
	Obese vs. normal	2.73 (1.21–6.19)	2.87 (1.00–8.19)
	Overweight/obese vs. normal	1.20 (0.69–2.05)	1.27 (0.73–2.20)

* adjusting for gender, parental marital status, age categories; § the results pertaining to the covariates of the model are shown in Table 3. OR: Odds ratios, Cl: confidence intervals.

**Table 3 children-08-00235-t003:** Odds ratios (OR) and 95% confidence intervals (Cl) for covariates assessed in the complex samples, multivariate logistic regression analysis examining the association between cyberbullying victimization, and overweight/obesity in all countries.

Covariates in the Overall Multivariate Analysis	Compared Categories	OR (95% CI)	*p*-Value
Gender	Male vs. female	0.75 (0.67–0.84)	<0.001
Age	16–17.9 vs. 14–15.9 years	1.25 (1.11–1.39)	<0.001
Parental marital status	Separated/divorced vs. married/living together	1.19 (1.06–1.35)	0.005

## Data Availability

The data presented in this study are available on request from the corresponding author.

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
