# Peer review of "Cyberbullying and Obesity in Adolescents: Prevalence and Associations in Seven European Countries of the EU NET ADB Survey"

_children, 2021, doi:10.3390/children8030235_

Round 1

Reviewer 1 Report

The topic is important and current. Please see my comments below:

-please check the affiliations, they look confusing. Contributions should be marked at the end of article.

ABSTRACT:

-please use decimals and parenthesis in a consistent way (throughout the article)

-was the situation also pronounced in Iceland as it was specified in the abstract?

INTRODUCTON:

-is reference missing for the lines 46-50?

-in line 63, is meta-analysis referring to 14 or 16 studies or something else, please clarify. Please check the dots at the end of sentences (throughout the article).

-what is meant by weight sigma? Should it be stigma.

-references are needed to state sentences "the association between overweight/obesity and cyberbullying has not been extensively examined" and "In light of the scarcity of studies"

MATERIALS AND METHODS:

-even though sampling and data collection procedures have been described elsewhere, it would be aproapriate to state why certain sampling method was used, and why ninth and tenth grade students were studied. Data is old as it was collected 2011-2012.

-if students were 9th and 10th graders, is it true that their age ranged between 14 and 18 years

-was parental consent required from all? even though the participant was 18years. What about adolescent informed consent.

-what were the themes (contents) of the questionnaire if it consisted of 52 items?

-The question defining cyberbullying in lines 112-119, is it based on reference/literature?

-sample size of the study can be appreciated, good!

RESULTS:

-sentences should not start with numbers

-please explain the results first and then refer to table. It is quite strange that the the whole paragraph starts with Table x...

-please delete Table 2 as all of it is explained in text

-marital status (table 3) of adolescents between 14-18 years of age were studied?? I am interested to know more about their marital status

-I suggest to include information in abstract that "No significant associations were identified either for obese or overweight adolescents in Greece, Spain, Romania, Poland and the Netherlands."

DISCUSSION:

-please synthesize your results and previous results

-time of data collection (almost 10 years ago) is limitation of the study. Why do the authors consider that it is still worth publishing.

CONCLUSIONS:

-what kind of causality authors suggest to study

-what measures prevent cyberbullying

Reviewer 2 Report

This study used a cross-sectional design to investigate the association between cyberbullying and overweight status in 7 adolescent populations. This is an ordinary study.  

Major comments:

  1. Because there have been several meta-analysis studies to evaluate the association between bullying and obesity status, what is the new and important for this study?
  2. This study was conducted with a complex sampling scheme, did and how the authors use a survey analysis module to take into the complex multistage survey account?
  3. Table 3, the association has been adjusted for ‘marital status’. Did the children have married? In this study, we cannot see the distribution about the confounder that the authors adjusted for.
  4. The authors should consider more about the bullying-associated factor as confounder, for example, the interpersonal relationship, individual personality, etc.
  5. The authors should offer the behavior mechanism or pathways for the relationship between overweight and cyberbullying.

Reviewer 3 Report

The article deals with the relationship between overweight/obesity and being a victim of cyberbullying among teenagers from several countries. After reading the manuscript, I would like to ask a few questions:

  1. Was the use of cyberbullying by the respondents assessed using one question out of 52 questions? Did this question completely explain that researchers want to find out how often someone has been subjected to cyberbullying and not used cyberbullying? It is possible that this was explained to the respondents, but it is not clearly written in the article, especially when you read this question.
  2. Were other information obtained in the study, apart from gender, age, and marital status? If so, why were they not included in the study?
  3. It should be clearly explained in the study (especially in places where conclusions are made) that it concerns situations when the respondents succumbed to cyberbullying, and not situations when they were used it.

Reviewer 4 Report

It should be noted that the topic of this study is very interesting and novel, considering that overweight and/or obesity is a public health problem, increasingly among young people. I believe that some aspects could be better nuanced, which I will detail below:

  1. Regarding lines 112-120, after reading the question used to define cyberbullyng, it does not make clear the answer that the participating adolescents should indicate. That is, the participants had to openly indicate if what had been suffered, a series of response options were available according to the frequency with which harassment occurred on the Internet or possibly indicate “yes or no”.
  2. From the data provided by Table 4, I am asked the following question. Is this same observed regardless of the country, or are these associations more marked in any of the participating countries?
  3. It must be recognized that the points used, percentiles, to determine overweight and obesity are dependent on the study population. Bearing in mind that the percentage of overweight or obesity in the different countries is not the same, has a sensitivity analysis been attempted using the percentiles depending on each country of origin?
  4. After reading the discussion, it is not clear to me the reasons why an association between cyberbullying and being overweight could be observed, but this association between cyberbullying and obesity is not so evident.
  5. The fact that the weight is self-reported could be a limitation, and I think that it should be considered in the last paragraph of the discussion. Although as a third limitation, mention is made of the self-report of cyberbullyng, in the same way, the weight is self-reported. It could happen that those subjects who are underweight or obese do not want to weigh themselves and the weight they report is distant in time or even that the weight they report is lower than the real one, for fear that their real weight is known.

Minor Comments

     6. In line 228 there is an "in" left over

Reviewer 5 Report

In this study, the authors reported the correlation between cyberbullying and obesity or overweight. As the authors put in the manuscript, one of the strengths of this study is the sample size, which consists of data collected from seven European countries. Suggestions are as below:

  1. In some countries, cyberbullying happened more frequently in overweight groups more than in obesity groups, what might be the underlying reasons? It is reasonable to presume that cyberbullying might be more severe in the obesity group than in the overweight group. 
  2. In the manuscript, the term "overweight" and "obesity" are not very clear. Sometimes it seems the authors are using "overweight" to indicate both "obesity" and "obesity". e.g.This study reveals an original association between cyberbullying and overweight on the basis of a sizable, representative sample of adolescent population from seven European countries. The association between overweight and cyberbullying was prominent in Germany, whereas an association with obesity emerged in Iceland. 
  3. In the discussion part, it seems that the authors aimed to discuss their results with the counterpart studies of other countries. However, results of Germany are listed in the part of Iceland. (Line 206-211).
  4. The methodology, which the authors employed to discuss their results, is confusing. Since counterpart results are discussed within countries, if possible, it might be better to discuss cyberbullying and traditional bulliyng at the same time.
  5. Since "overweight" and "obesity" earned different pattern of results in different country, it might be a better way to combine the "overweight" and "obesity" together, since they both represent "abnormal, excessive body weight". 
  6. Data showed in Table 4 was not discussed extensively. 
  7. In the conclusion, the authors wrote: The present findings are interesting from a preventive and therapeutic perspective; schools can apply interventions and take measures in order to prevent cyberbullying towards overweight and obese adolescents. What are the potential therapeutic and perspectvie are ? They are not presented throughout the manuscript.
  8. As the authors wrote in the manuscript, data presented here is only part of a large study, if possible the prevalence of cyberbullying in adolescents should be analyzed with other applicable data.

Round 2

Reviewer 2 Report

Some issues are unclear and need to be clarified, as the follows:

  1. Abstract, lines 32-33. Conclusion: No significant associations were identified… What type association was implied? This sentence is unclear.

  1. With regard to “Point 4: The authors should consider more about the bullying-associated factor as confounder, for example, the interpersonal relationship, individual personality, etc.” The authors responded that this study is a part of a large study, thus they should have the data on interpersonal relationship and individual personality. I suggested to adjust for the effect of these variables in the related results.

  1. With regard to “Point 5: The authors should offer the behavior mechanism or pathways for the relationship between overweight and cyberbullying.” The authors provided sociological theories to explain the association between cyberbullying and overweight status. However, the statement is still unclear. Alternatively, as we have known, the major factors associated with overweight/obesity are genetic characteristics, dietary intakes, and physical activity. How can we ascertain the observed association in this study was not the results derived from these factors?

Reviewer 4 Report

Thanks for the clarifications and changes made. I believe that the article could be published in its current form.

Reviewer 5 Report

Thank you for your responses. 

I believe the quality of the manuscript has been improved.
